# Sponsored Question Answering

## ABSTRACT

The potential move from search to question answering (QA) ignited the question of how should the move from sponsored search to sponsored QA look like. We present the first formal analysis of a sponsored QA platform. The platform fuses an organic answer to a question with an ad to produce a so called *sponsored answer*. Advertisers then bid on their sponsored answers. Inspired by Generalized Second Price Auctions (GSPs), the QA platform selects the winning advertiser, sets the payment she pays, and shows the user the sponsored answer. We prove an array of results. For example, advertisers are incentivized to be truthful in their bids; i.e., set them to their true value of the sponsored answer. The resultant setting is stable with properties of VCG auctions.

## 1 INTRODUCTION

In sponsored search [9], advertisements (henceforth referred to as ads) are ranked in response to a query. A click on an ad usually leads to a landing page on the Web. In contrast to organic search, the ranking is not based on relevance estimation but rather on monetization criteria. For example, by the foundational General Second Price (GSP) Auction [7, 18], ads are ranked by bids posted by the advertisers[1] where the payment of an advertiser is the bid posted by the advertiser ranked just below her. More evolved versions include measures of ad quality (e.g., predicted click through rate) in the ranking criterion [7, 18].

Recently, there is an on-going discussion, and some implementations, of using question answering systems as potential alternatives to search engines [2, 4, 12]. This is due to the dramatic progress with large language models (LLMs) [21]. An emerging question is then how sponsored search will evolve with the transition from search to question answering. A few recent suggestions include showing ads during a QA conversation (chat), providing the ad as an answer in a QA session, and integrating an answer with an ad [1, 4, 8].

However, to the best of our knowledge, there is still no theoretical treatment of potential mechanisms for sponsored QA. In this paper, we make a first step towards this end. We present a formal auction mechanism for a sponsored QA platform. The use of ads in the QA platform we study is inspired by Feizi's et al. [8] conceptual proposal and by Microsoft's alleged approach [15] to fuse (integrate) answers with ads. For example, suppose the question *"Which book is*

---

[1] Advertisers bid on keywords. If a keyword appears in a query then the ad is candidate for ranking.

*used in most CS programs in the U.S. to teach Python?"* has an answer *"Book X"* generated by some LLM. Suppose there is an ad about a specific bookstore Y: *"Bookstore Y has all the programming books one might need to thoroughly learn how to program. For example, we highly recommend book X which consistently receives excellent reviews."*. The result of fusing the answer and the ad can be: *"In bookstore Y you can find book X which is used by most CS programs in the U.S. to teach Python. The book consistently receives excellent reviews."*

Our suggested QA platform operates as follows. Given a user question, the platform generates an *organic answer* which is independent of monetization considerations, e.g., using an LLM. The platform fuses the answer with each of the candidate ads to produce so called *sponsored answers*. Then, each advertiser places a bid for her sponsored answer. Inspired by the GSP auctions used in sponsored search [18], we devise a criterion for selecting the sponsored answer and the payment the selected advertiser should pay. We prove that a *dominant strategy* of advertisers is to place a bid equal to the value of the sponsored answer for them. This result which is a property of VCG auctions [11] is important since the auction setting is in a stable (equilibrium) state: advertisers are incentivized to be truthful in their bids rather than engage in *shading* [14] — i.e., continuously changing bids to improve gains.

We further define a notion of *social welfare* which is an aggregate of the value of a sponsored answer to the advertiser and the user utility attained by this answer. Then, in the spirit of VCG auctions [11], we show that under the dominant strategy, the utility of the winning advertiser (defined as the difference between the value and the payment) is the difference between the social welfare given her sponsored answer and the social welfare given the sponsored answer of the second best advertiser whose bid is used to determine the winner's payment.

Our formal analysis is agnostic to the fusion approach employed to integrate the organic answer and an ad. We discuss a few potential fusion methods using large language models (LLMs). However, formally coupling the result of LLM-based fusion and the auction mechanism is extremely difficult to impossible. Hence, to provide an end-to-end formal analysis of the process that starts with fusion and continues with the auction, we use unigram language models. We formally show that the advertiser with the highest bid is not necessarily the winner and that the winning sponsored result does not necessarily maximize user utility. The latter result can be attributed to the fact that we devise the platform to maximize both users' and advertisers' satisfaction.

To summarize, our main contributions are (i) presenting the first — to the best of our knowledge — formal proposal of a sponsored QA system which is based on an auction mechanism, and (ii) proving a few important theoretical results about the platform.

## 2 RELATED WORK

We assume that the QA platform generates an organic answer to a question, e.g., using a large language model (LLM). Indeed, question answering is a prominent application of using large language models [21]. We note that answer generation is not the focus of our work here. Any LLM can be used to generate the organic answer in our suggested platform.

Sponsored search is central to the monetization of search engines [3]. In sponsored search, ads are ranked in response to a query. In contrast, following recent proposals [8, 15], we assume that an ad is fused with an organic answer to a question.

Two prevalent auctions on the Web are VCG [11] and Generalized Second Price (GSP) auctions [7, 18]. These auctions are equivalent when a single advertisement is selected [7, 19]. In this case, the highest bidding advertiser wins, and the payment is the minimum bid necessary to secure the win, specifically, the bid of the advertiser with the second highest bid. Our suggested auction is adaptation of a GSP auction to the sponsored QA setting. Since our QA platform strives to satisfy both users and advertisers, the winning advertiser is not necessarily the one who posted the highest bid as we show.

There is work on content generation using multiple LLMs where an incentive-compatible auction[2] is held for *each* generated token [6]. Our platform is fundamentally different as bids are posted for a given question and a sponsored answer created from an ad and an organic answer. Furthermore, in contrast to our work, there is strong coupling between the content generation approach and the auction [6]. Our platform is agnostic to the QA algorithm used to generate an organic answer and the approach used to fuse it with an ad. Furthermore, in contrast to Duetting et al. [6], we define and analyze social welfare.

Dubei et al. [5] devise an auction mechanism where ads of advertisers are summarized using an LLM. The advertisers essentially bid on the prominence of their ad in the summary. In contrast, in our QA platform a single sponsored answer created from an ad and an organic answer is presented to the user. Hence, the auction mechanisms are completely different and so is the corresponding analysis.

Recently, Feizi et al. [8] proposed a conceptual framework for online advertising in LLM-based QA systems. They discuss important considerations including privacy, latency, reliability, users' and advertisers satisfaction. However, no formal/algorithmic framework is proposed in their work. We adopt Feizi et al.'s [8] proposal of fusing an organic answer with an ad which is also allegedly the case in some implementations [15]. Our focus in this paper is on devising an auction mechanism and analyzing it using several perspectives. The important issues discussed by Feizi et al. [8] are outside the scope of this paper, but certainly deserve in-depth future exploration. A case in point, the reliability of sponsored answers generated from organic answers and ads is highly important.

## 3 SPONSORED QUESTION ANSWERING PLATFORM

We now turn to present a question answering (QA) platform which responds to users' questions with answers which include sponsored information.

Suppose that a user posts a question $q$ and the QA platform generates an *organic* (textual) answer $d_o$ which is not affected by monetization considerations. There are $n$ advertisers (content providers) interested in presenting their *textual* advertisements (ads in short) in response to $q$.[3] We assume that each advertiser $i$ ($\in N := \{1, 2, ..., n\}$) has a single ad $d_i$ she wants to present. The platform fuses for every advertiser $i$ her ad ($d_i$) with the organic answer $d_o$ to produce a *sponsored answer* $d_i^s$ which will be shown to the user in case advertiser $i$ is selected. The fusion process can be advertiser-specific and be based on commercial terms between the platform and the advertiser. A case in point, the relative emphasis on the organic answer versus that on the ad can be the result of a commercial agreement. The platform then runs an auction in which the advertisers bid on the corresponding sponsored answers. The platform uses the bids with additional information to determine the winner of the auction and her payment.

Since auctions are essentially games composed of players (bidders) and their strategies (bids) we start by describing in Section 3.1 some basic game theory concepts. In Section 3.2 we formalize the auction mechanism and present some theoretical results. The formalism and results are agnostic to the actual fusion approach employed to fuse an organic answer and an ad. In Section 4 we discuss two options of fusion using large language models, and in addition provide a theoretical analysis of the results of using unigram language models for fusion.

### 3.1 Game Theory

We now briefly review some basic concepts in game theory.

DEFINITION 1. *An n-players game is a tuple* $G = (\{S_i\}_{i \in N}, \{U_i\}_{i \in N})$. *$S_i$ is the set of strategies of player $i$. $S = S_1 \times S_2 \times ... \times S_n$ is the set of **strategy profiles** in the game. $U_i : S \to \mathbb{R}_+$ is the **utility function** of player $i$.[4]*

Each player in a game aims to maximize her utility. Note that the utility depends not only on her strategy but also on the strategies of other players. Consider the following 2-players game: $S_1 = S_2 = [0, 1]$, $U_1(s_1, s_2) = 1 + s_1^2 + s_2^2$, $U_2(s_1, s_2) = 2 + s_1^2 + s_2^2$. If the players select $s_1 = 1$ and $s_2 = 1$, then their utilities are $U_1(1, 1) = 3$ and $U_2(1, 1) = 4$.

A fundamental characterization of games is their stability or lack thereof. A *stable* strategy profile of a game is a profile where no player has an incentive to deviate from her strategy. A well known example of a stable profile is Nash equilibrium. Let $S_{-i} = S_1 \times ... \times S_{i-1} \times S_{i+1} \times ... \times S_n$ denote the strategy profile of all the players except $i$. Then,

DEFINITION 2. *A strategy profile $s = (s_i, s_{-i})$ where $s_i \in S_i$, $s_{-i} \in S_{-i}$ is a Nash equilibrium if $\forall i \in N$ and $\forall s_i' \in S_i$, $U_i(s_i', s_{-i}) \leq U_i(s_i, s_{-i})$.[5]*

Considering the game example from above, it is easy to show that the strategy profile $s_1 = 1$, $s_2 = 1$ is a Nash equilibrium.

A *dominant strategy* of a player is a strategy that is better than any other strategy regardless of the strategies of other players:

---

[2]In these auctions, bidders are incentivized to bid their true value.

[3]The advertisements can include images and video captions, but henceforth we focus on the textual part,

[4]$\mathbb{R}_+$ is the set of non-negative real numbers.

[5]We focus on pure strategies and pure Nash equilibrium. Mixed strategies which are distributions over pure strategies are outside the scope of this paper.

DEFINITION 3. $s_i \in S_i$ *is a* dominant strategy *of player $i$ if $\forall s_{-i} \in S_{-i}, \forall s_i' \in S_i, U_i(s_i', s_{-i}) \leq U_i(s_i, s_{-i})$.*

Note that players' strategies in a Nash equilibrium (Definition 2) need not necessarily be dominant strategies. In our example game, $s_1 = 1$ is a dominant strategy of the first player, since $U_1(1, s_2) \geq U_1(s_1, s_2) \ \forall s_1, s_2 \in [0, 1]$.

A common measure of the "goodness" of a strategy profile $s$ is *social welfare*, often defined as the sum of the players' utilities:

$$SW(s) = \sum_{i \in N} U_i(s). \tag{1}$$

## 3.2 Auctions for Sponsored Answers

We next turn to formalize the auction employed by the QA platform. Recall that prior to the auction, each ad $d_i$ is fused with the organic answer $d_o$ to yield a sponsored answer $d_i^s$. Advertiser $i$ then places a bid for $d_i^s$ to be shown. The platform selects the sponsored answer to show and determines the payment of the winning advertiser.

DEFINITION 4. *A* question-specific setup *of a question-answering (QA) platform with sponsored answers is a tuple $QAS := (q, d_o, N, \{d_i\}_{i \in N}, \{d_i^s\}_{i \in N}, \{v_i(d_i^s)\}_{i \in N})$. $q$ and $d_o$ are a question and the organic answer generated for it, respectively. $N := \{1, 2, ..., n\}$ is a set of advertisers where $n \geq 2$. $\{d_i\}_{i \in N}$ are the advertisers' ads for $q$. $d_i^s$ is the sponsored answer generated by fusing the organic answer $d_o$ with ad $d_i$. $v_i(d_i^s) \in \mathbb{R}_+$ is the value for advertiser $i$ of displaying the sponsored answer $d_i^s$ to the user.*

We assume the QA platform has to select a single sponsored answer from $\{d_i^s\}_{i \in N}$ to show the user. This practice is aligned with a setting where the platform shows a single organic answer. As in classical sponsored search [9], the platform is incentivized to receive a payment based on the content it presents to users. We now turn to define the utilities of the stakeholders — users, advertisers and the platform itself — and propose an auction mechanism to determine the winner and her payment.

We focus on a setup where the (game) strategy $s_i$ of every advertiser $i$ is a bid $b_i(d_i^s)$ ($\in \mathbb{R}_+$) for showing the sponsored answer $d_i^s$ to the user. The set of *strategy profiles* of the advertisers is $S = S_1 \times S_2 \times ... \times S_n$ where $S_i$ is the set of all possible bids of advertiser $i$; namely, $\mathbb{R}_+$. We adapt GSP (generalized second price auction) [18] to our setting for the allocation and payment procedures.

The platform's value for a sponsored answer $d_i^s$ is an aggregation of the user utility and the advertiser's bid on $d_i^s$:

DEFINITION 5. *The prospect* Platform Value *(henceforth, platform value in short) from displaying the sponsored answer $d_i^s$ is defined as $PV(d_i^s, q, b_i(d_i^s)) = U(d_i^s, q) + b_i(d_i^s)$, where $U(d_i^s, q)$ is the user utility attained by seeing the sponsored answer $d_i^s$ in response to her question ($q$), and $b_i(d_i^s)$ is the bid of the advertiser $i$ on the sponsored answer $d_i^s$.*

We use the term "prospect" to emphasize that the payment received by the platform can be lower than $b_i(d_i^s)$ as explained below. We next define social welfare with respect to a sponsored answer as the sum of the user utility and the value for the advertiser whose sponsored answer was selected. In standard GSPs for sponsored search [18], the social welfare is a weighted sum of advertisers'

values (cf., Equation 1) where the weights represent click probabilities[6]. Here we do not account for click probabilities. We do take into account, in contrast to standard GSPs, the user's utility.

DEFINITION 6. *The* Social Welfare *from displaying the sponsored answer $d_i^s$ is $SW(d_i^s, q, b_i(d_i^s)) = U(d_i^s, q) + v_i(d_i^s)$.*

Note that the social welfare is composed of a user part (utility) and advertiser part (value). Indeed, the platform aims at satisfying both users and advertisers.

The advertiser selected as the *winner* of the auction, $i^w$, is the one whose sponsored answer and bid maximize the platform value:

$$i^w := arg \max_i PV(d_i^s, q, b_i(d_i^s)). \tag{2}$$

The platform shows the sponsored answer $d_{i^w}^s$.

In GSP auctions [18], the winner pays the minimal payment $p_{i^w}(d_{i^w}^s) \leq b_{i^w}(d_{i^w}^s)$ needed to surpass the second best platform value, attained by advertiser $i^{snd}$. Formally, $PV(d_{i^{snd}}^s, q, b_{i^{snd}}(d_{i^{snd}}^s)) \leq PV(d_{i^w}^s, q, b_{i^w}(d_{i^w}^s))$ and $PV(d_{i^{snd}}^s, q, b_{i^{snd}}(d_{i^{snd}}^s)) = PV(d_{i^w}^s, q, p_{i^w}(d_{i^w}^s))$. Based on these definitions, we arrive to the following result:

LEMMA 1. *The payment $p_{i^w}(d_{i^w}^s)$ of the winner $i^w$ is $b_{i^{snd}}(d_{i^{snd}}^s) + U(d_{i^{snd}}^s, q) - U(d_{i^w}^s, q)$.*

PROOF. The winner $i^w$ pays the minimal $p_{i^w}(d_{i^w}^s)$ s.t. the resultant platform value is the second best. Since $PV(d_{i^{snd}}^s, q, b_{i^{snd}}(d_{i^{snd}}^s)) = PV(d_{i^w}^s, q, p_{i^w}(d_{i^w}^s))$, then $U(d_{i^{snd}}^s, q) + b_{i^{snd}}(d_{i^{snd}}^s) = U(d_{i^w}^s, q) + p_{i^w}(d_{i^w}^s)$ and therefore $p_{i^w}(d_{i^w}^s) = b_{i^{snd}}(d_{i^{snd}}^s) + U(d_{i^{snd}}^s, q) - U(d_{i^w}^s, q)$. □

We now define the utilities of advertisers. Note that the utility of an advertiser depends on the bids of all other advertisers, not only her own; specifically, the payment of the winner depends on the bid of another advertiser. Indeed, recall from Section 3.1 that the utility of a player in a game depends on the strategies employed by all players.

Following common practice in auction theory [13], we assume quasi-linear utility functions of the advertisers: we substract the payment from the value to determine the advertiser utility if she is the winner, and set the utility to 0 otherwise.

DEFINITION 7. *The utility of advertiser $i$ is:*

$$U_i^a(d_i^s, v_i(d_i^s), \{d_j^s\}_{j \neq i}, \{v_j(d_j^s)\}_{j \neq i}) = \begin{cases} v_i(d_i^s) - p_i(d_i^s) & \text{if } i \text{ wins,} \\ 0 & \text{otherwise;} \end{cases} \tag{3}$$

*$p_i(d_i^s)$ is $i$'s payment in case she wins.*

The auction setting we defined can be analyzed as a game where the advertisers' strategies are their bids and they strive to maximize their utility. We next show that advertisers have an incentive to behave truthfully, that is, bid the true value of their sponsored answer. More specifically, this bidding strategy is a dominant strategy. As a result, the game (auction) has a Nash equilibrium (i.e., the game is stable). This stability is highly important. A case in point, in auctions which are not incentive-compatible (i.e, bidders

---

[6]In standard GSPs for sponsored search, ads are ranked. In our setting, a single ad is presented, and only its advertiser has potential gain.

are not incentived to bid their true value) *shading* phenomena [14] are prevalent: bidders continuously addapt their bids (specifically, lower than the true value) to maximize gain.

LEMMA 2. $b_i(d_i^s) = v_i(d_i^s)$ *is a dominant strategy of advertiser* $i$.

PROOF. We show that the utility of player $i$ is maximized when $b_i(d_i^s) = v_i(d_i^s)$. Let $M = max_{j \neq i} PV(d_j^s, q, b_j(d_j^s))$ be the maximal platform value attained by selecting a player $i' \neq i$. There are two options for the utility of advertiser $i$:

- $i$ is not the winner: $PV(b_i(d_i^s)) \leq M$. In this case, the utility of player $i$ is $U_i(b_i(d_i^s)) = 0$ and $b_i(d_i^s) = v_i(d_i^s)$ is a dominant strategy.
- $i$ is the winner: $PV(b_i(d_i^s)) > M$. Hence, $j = i^{snd}$ and $M = b_{i^{snd}}(d_{i^{snd}}^s) + U(d_{i^{snd}}^s, q)$. By Lemma 1, $p_i(d_i^s) = M - U(d_i^s, q)$. The utility of $i$ is then $v_i(d_i^s) - p_i(d_i^s) = v_i(d_i^s) - M + U(d_i^s, q)$. We analyze two cases :
  - if $v_i(d_i^s) - M + U(d_i^s, q) < 0$ then $v_i(d_i^s) < U(d_{i^{snd}}^s, q) + b_{i^{snd}}(d_{i^{snd}}^s) - U(d_i^s, q)$ and $v_i(d_i^s) + U(d_i^s, q) < b_{i^{snd}}(d_{i^{snd}}^s) + U(d_{i^{snd}}^s, q)$. If $b_i(d_i^s) \leq v_i(d_i^s)$, we get $b_i(d_i^s) + U(d_i^s, q) < b_{i^{snd}}(d_{i^{snd}}^s) + U(d_{i^{snd}}^s, q)$. Consequently, $i^{snd}$ is the winner and we get a contradiction.
  - if $v_i(d_i^s) - M + U(d_i^s, q) \geq 0$, then $i$ can maximize her utility by winning the game having $v_i(d_i^s) - p_i(d_i^s) \geq 0$; to this end, $i$ can bid $b_i(d_i^s) = v_i(d_i^s)$ since $v_i(d_i^s) + U(d_i^s, q) \geq M$. □

It is easy to show that the utility of the winner is the difference between the social welfare if her sponsored answer is selected and the social welfare if the sponsored answer of the second best advertiser (in terms of platform value) is selected. This is a property of VCG auctions [11]. Formally,

PROPOSITION 1. *Assume all advertisers play their the dominant strategy:* $b_i(d_i^s) = v_i(d_i^s)$. *The utility of the winner* $i^w$ *is:* $v_{i^w}(d_{i^w}^s) - v_{i^{snd}}(d_{i^{snd}}^s) + U(d_{i^w}^s, q) - U(d_{i^{snd}}^s, q) = SW(d_{i^w}^s, q, b_{i^w}(d_{i^w}^s)) - SW(d_{i^{snd}}^s, q, b_{i^{snd}}(d_{i^{snd}}^s))$.

## 4 FROM ADS TO SPONSORED ANSWERS

Heretofore, we treated the fusion of the organic answer $d_o$ and the ad $d_i$ to yield a sponsored result, $d_i^s$, as a black box. That is, advertiser $i$ bids on $d_i^s$ based on the value $v_i(d_i^s)$ it attributes to it. Then, the QA platform selects the sponsored answer that yields the highest platform value (see Definition 5). We now turn to describe possible fusion methods and discuss their implications.

### 4.1 Sponsored Answer Generation Using Large Language Models

A possible approach to fusing the organic answer $d_o$ and an ad $d_i$ is to request (via a prompt) a large language model (LLM; e.g., GPT) to perform the fusion. However, it is impossible to make theoretical statements about the resultant sponsored answer, $d_i^s$.

Furthermore, it is highly difficult, to impossible, to quantitatively control the relative emphasis on $d_o$ versus $d_i$ in a prompting-based fusion approach as just described. This aspect can be quite important for both the advertiser and the user of the QA platform: the advertiser strives to have the sponsored answer $d_i^s$ similar to $d_i$ and

the user probably wants it to be similar to the organic answer, $d_o$. We re-visit this point below.

We now consider an alternative approach to fusion inspired by a recent proposal of integrating multiple LLMs for content generation [6]. Suppose advertiser $i$ generates the (textual) ad $d_i$ using an LLM, denoted $\theta^{ad}$. Suppose also that the QA platform uses an LLM, $\theta^{org}$, to generate the organic answer $d_o$ for the question $q$ which is used as the prompt. Assume that both LLMs are autoregressive. That is, given a sequence of tokens $x$ (composed of the prompt and tokens already generated), there is a probability distribution over the token vocabulary $V$ from which the next token is sampled. Formally, $p(t|\theta, x)$ is the probability of token $t$ given the LLM $\theta$ and the sequence $x$ already generated. Then, we can fuse the organic answer $d_o$ and the ad $d_i$ by (i) defining a next-token generation distribution: $\lambda p(t|\theta^{org}, x) + (1-\lambda)p(t|\theta^{ad}, x)$, where $\lambda$ is a parameter controlling the relative emphasis on the organic answer and the ad; and (ii) using the original prompts in each of the two LLMs: the request to generate the ad ($\theta^{ad}$) and the question ($\theta^{org}$).

There are three main challenges embodied in the fusion process just described. First, there is no guarantee about the quality of the generated sponsored answer, $d_i^s$, since we mix distributions at the token level. Second, we assumed that an ad was generated using an LLM which is not necessarily the case. Third, we still cannot theoretically reason about the generated sponsored answer and formally use it in the advertiser value function and user utility function defined above.

To facilitate a first step towards a formal end-to-end (theoretical) analysis of the process of fusing $d_o$ and $d_i$ to produce $d_i^s$, and using $d_i^s$ in the auction defined in Section 3.2, we subscribe to the language modeling framework to retrieval [10, 16]. Specifically, we use unigram language models based on a multinomial distribution [10, 16, 17].

### 4.2 Unigram Language Models

Let $x$ be a text and $t$ a term (token) in the vocabulary $V$. $p_x(t) := \frac{tf(t,x)}{\sum_{t' \in x} tf(t',x)}$ is the maximum likelihood estimate of $t$ with respect to $x$ assuming a multinomial distribution; $tf(t, x)$ is the number of times $t$ appears in $x$. $p_x(\cdot)$ is the unsmoothed unigram language model induced from $x$. The probability assigned to a term sequence $t_1, \ldots, t_n$ is $p_x(t_1, \ldots, t_n) := \prod_{i=1}^{n} p_x(t_i)$. Often, unigram language models are smoothed to avoid the zero probability problem [20]. For our formal analysis herein, smoothing is not required. We hasten to point out however that our findings also hold for smoothed language models.

Given an organic answer $d_o$ and an ad $d_i$, we can fuse the unigram language models induced from them using a linear mixture to produce a unigram language model from which a sponsored answer will be generated: $p_i^{spon}(t) := \lambda_i p_{d_o}(t) + (1 - \lambda_i)p_{d_i}(t)$. Note that $\lambda_i$ is advertiser-specific and can potentially be negotiated between the advertiser and the QA platform before the auction takes place. We can now generate a specific sponsored answer $d_i^s$ of length $k$ by sampling $k$ times from $p_i^{spon}(\cdot)$. The *mean* number of occurrences of token $t$ in a document of length $n$ generated using $p_i^{spon}(\cdot)$, i.e., in a sponsored answer, is $np_i^{spon}(t)$.[7] Herein, we use

---

[7]This is the mean in a multinomial distribution.

$d_i^s$ to denote the "mean" document which is composed of these mean number of occurrences of tokens, henceforth simply referred to as the sponsored answer. As was the case of fusing LLMs at the next-token generation level, there is no guarantee about the quality of the generated sponsored answer. We hasten to point out that our use of unigram language models is intended to facilitate the formal analysis of the auction mechanism given a specific fusion method.

**Advertiser's value, user's utility and text similarity**. We recall that advertiser $i$ is interested in having her ad $d_i$ presented for the question $q$. Instead, the platform suggests showing the sponsored answer $d_i^s$ which is the result of fusing $d_i$ with the organic answer $d_o$. Naturally, the advertiser strives to have the sponsored answer as similar as possible to her ad. Hence, we define the advertiser's value function used in Definition 5 as

$$v_i(d_i^s) := S(d_i^s, d_i), \tag{4}$$

where $S$ is an inter-text similarity measure.

Similarly, we assume that the goal of the user who posted the question $q$ so as to (presumably) receive an organic answer is to have the sponsored answer $d_i^s$ as similar as possible to the organic answer $d_o$. Accordingly, we define the user utility function used in Definition 5 as

$$U(d_i^s, q) := S(d_i^s, d_o). \tag{5}$$

We define the similarity of texts $x$ and $y$ based on the cross entropy measure (cf. [10]):

$$S(x, y) := 2A - CE(p_x(\cdot)||p_y(\cdot)) - CE(p_y(\cdot)||p_x(\cdot)); \tag{6}$$

$CE(p_x(\cdot)||p_y(\cdot)) = \sum_{t \in V} p_x(t) log p_y(t)$;[8] lower cross entropy corresponds to increased similarity[9]; $A \in \mathbb{R}_+$ is used to ensure that the similarity value is in $\mathbb{R}_+$: $A := \max_{j_1, j_2} CE(p_{d_{j_1}}(\cdot)||p_{d_{j_2}}(\cdot)) + CE(p_{d_{j_2}}(\cdot)||p_{d_{j_1}}(\cdot))$ where $d_{j_1}, d_{j_2} \in \{d_o\} \cup \{d_i\}_{i \in N}$.

**Auction Analysis with Unigram Language Models**. We now turn to show that in the unigram-language-model setting described above, the advertiser $i$ who wins the auction is not necessarily the one with the maximal value $v_i(d_i^s)$. Since by Lemma 2 the dominant strategy is to set the bid as the value, we arrive to an interesting result: the advertiser who placed the highest bid is not necessarily the one who wins the auction. This result is in contrast to the state-of-affairs in many other auction mechanisms, and is due to the QA platform's goal to satisfy both users and advertisers.

PROPOSITION 2. *The winner of the auction is not necessarily advertiser $i$ whose value $v_i(d_i^s)$ is the maximum with respect to all advertisers' values. Accordingly, the winner is not necessarily the advertiser who placed the highest bid.*

PROOF. Consider a 2-advertisers setting with a two-terms vocabulary: $V = \{a, b\}$. Suppose $d_o, d_1$ and $d_2$ are all sequences of $k$ terms. Suppose $(1-\epsilon)$ of the tokens in $d_o$ are $a$ and $\epsilon$ are $b$ ($\epsilon \in [0, 1]$). Consequently: $p_{d_o}(a) = 1-\epsilon$ and $p_{d_o}(b) = \epsilon$. For $d_1$ we assume: $p_{d_1}(a) = \epsilon$ and $p_{d_1}(b) = 1 - \epsilon$. For $d_2$ we assume: $p_{d_2}(a) = p_{d_2}(b) = 0.5$.

---

[8]We use cross entropy rather than KL divergence because the cross entropy is linear in its left argument. Note that the resultant similarity function is concave.

[9]As noted above, usually language models are smoothed so as to avoid zero probabilities (e.g., in cross entropy computation). For our purposes (constructive proofs presented below), smoothing is not needed as there are no cases of zero probabilities.

Following the fusion approach described above, $p_{d_i}^s(t) = \lambda_i p_{d_o}(t) + (1 - \lambda_i) p_{d_i}(t)$ for $i \in \{1, 2\}$ and $t \in \{a, b\}$. We set $\lambda_1 = \epsilon$ and $\lambda_2 = 1 - \epsilon$. It can be shown that for $\epsilon \to 0$, $S(d_1^s, d_1) = v_1(d_1^s) > S(d_2^s, d_2) = v_2(d_2^s)$ and $v_2(d_2^s) + S(d_2^s, d_0) > v_1(d_1^s) + S(d_1^s, d_0)$; i.e., $v_2(d_2^s) + U(d_2^s, q) > v_1(d_1^s) + U(d_1^s, q)$. Thus, $d_2$ wins the auction but the value for $d_1$ is higher. The full proof is provided in Appendix A.1. Since by Lemma 2 bidding the value is a dominant strategy, we get that the winner of the auction is not the one who placed the highest bid. □

We next show that the user utility is not necessarily the maximal possible. This is, again, due to the fact that the QA platform aims to satisfy both users and advertisers.

PROPOSITION 3. *If advertiser $i$ won the auction, it is not necessarily the case that the resultant user utility, $U(d_i^s, q)$, is the maximal with respect to that attained by selecting other advertisers (sponsored answers).*

PROOF. We consider the same 2-advertisers setting as in Proposition 2 with the same ads and organic answer. The difference is that now we set $\lambda_1 = 1 - \epsilon$ and $\lambda_2 = 0.5$.

It can be shown that for $\epsilon \to 0$, $U(d_1^s, q) = S(d_1^s, d_o) > S(d_2^s, d_o) = U(d_2^s, q)$ and $v_2(d_2^s) + U(d_2^s, q) > v_1(d_1^s) + U(d_1^s, q)$. That is, $i = 2$ is the winner of the auction but the resultant user utility is lower than that for advertiser 1. The full proof is provided in Appendix A.2. □

# 5 CONCLUSIONS AND FUTURE WORK

We presented a novel formal platform for sponsored question answering (QA). The platform is based on fusing an organic answer to a question with an ad so as to produce a sponsored answer. Advertisers bid on their corresponding sponsored answers. Inspired by principles of Generalized Second Price Auctions (GSPs) [7, 18], the platform selects the sponsored answer to show the user and sets the payment for the selected advertiser.

We prove that a dominant strategy of advertisers is to bid on their true value of the sponsored answer which is a property of VCG [11] auctions. The result is that the QA setting reaches a stable state (equilibrium) where advertisers have no incentive to continuously change their bids (a.k.a., shading). We also formalize the notion of social welfare and show that the utility of the advertiser who wins the auction is the difference between the social welfare attained when presenting her sponsored answer and the social welfare of presenting the sponsored answer of the second best advertiser whose bid is the payment the winner has to pay.

Our general analysis of the auction is not committed to a specific approach to fusing an organic answer and an ad. To theoretically analyze end-to-end the process of fusing an organic answer with an ad and apply the auction, we use unigram language models. We prove that the auction winner is not necessarily that with the highest bid. We also prove that the attained user utility is not the maximal possible with respect to selecting other advertisers. This result is due to the fact that we design the QA platform to satisfy both users and advertisers.

In the unigram-language-model analysis, we used an inter-text similarity measure as a basis for defining advertisers' value functions and users' utility functions. For future work we plan to study the effect of using alternative value and utility functions.

The task of fusing an organic question with an ad deserves an in-depth future study. Such a study calls for the creation of evaluation datasets.

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

## A PROOFS

In what follows $CE(d_a||d_b)$ stands for $CE(p_{d_a}(\cdot)||p_{d_b}(\cdot))$. In both proofs below, we omit in the computations factors which are $O(\epsilon)$ or $O(\epsilon \log \epsilon)$ since we assume $\epsilon \to 0$.

## A.1 Proof of Proposition 2

Recall that $p_{d_o}(a) = 1 - \epsilon, p_{d_o}(b) = \epsilon, p_{d_1}(a) = \epsilon, p_{d_1}(b) = 1 - \epsilon$, $p_{d_2}(a) = 0.5, p_{d_2}(b) = 0.5$. Since $\lambda_1 = \epsilon, \lambda_2 = 1 - \epsilon$:

$p_{d_1^s}(a) = \lambda_1 p_{d_o}(a) + (1-\lambda_1)p_{d_1}(a) = \epsilon(1-\epsilon) + (1-\epsilon)\epsilon = 2\epsilon(1-\epsilon)$.

$p_{d_1^s}(b) = \lambda_1 p_{d_o}(b) + (1 - \lambda_1)p_{d_1}(b) = \epsilon \cdot \epsilon + (1 - \epsilon)(1 - \epsilon) = \epsilon^2 + (1 - \epsilon)^2$.

$p_{d_2^s}(a) = \lambda_2 p_{d_o}(a) + (1 - \lambda_2)p_{d_2}(a) = (1 - \epsilon)(1 - \epsilon) + \epsilon \cdot 0.5 = (1 - \epsilon)^2 + \frac{\epsilon}{2}$.

$p_{d_2^s}(b) = \lambda_2 p_{d_o}(b) + (1 - \lambda_2)p_{d_2}(b) = (1 - \epsilon) \cdot \epsilon + \epsilon \cdot 0.5 = \epsilon \cdot (1 - \epsilon) + \frac{\epsilon}{2}$.

We now compute $v_1(d_1^s), v_2(d_2^s), U(d_1^s, q), U(d_2^s, q)$ using the fact that $CE$ is linear in its left argument:

$v_1(d_1^s) = S(d_1^s, d_1) = 2A - CE(d_1^s||d_1) - CE(d_1||d_1^s) = 2A - \epsilon CE(d_o||d_1) - (1 - \epsilon)CE(d_1||d_1) + \epsilon log(2\epsilon(1 - \epsilon)) + (1 - \epsilon)log((1 - \epsilon)^2 + \epsilon^2) = 2A - CE(d_1||d_1)$.

$v_2(d_2^s) = S(d_2^s, d_2) = 2A - CE(d_2^s||d_2) - CE(d_2||d_2^s) = 2A - (1 - \epsilon)CE(d_o||d_2) - \epsilon CE(d_2||d_2) + 0.5log((1 - \epsilon)^2 + \frac{\epsilon}{2}) + 0.5log(\epsilon(1 - \epsilon) + \frac{\epsilon}{2}) = 2A - CE(d_o||d_2) + 0.5log(\frac{3\epsilon}{2} - \epsilon^2)$.

$U(d_1^s, q) = S(d_1^s, d_o) = 2A - CE(d_1^s||d_o) - CE(d_o||d_1^s) = 2A - \epsilon CE(d_o||d_o) - (1 - \epsilon)CE(d_1||d_o) + (1 - \epsilon)log(2\epsilon(1 - \epsilon)) + \epsilon log((1 - \epsilon)^2 + \epsilon^2) = 2A - CE(d_1||d_o) + log(2\epsilon(1 - \epsilon))$.

$U(d_2^s, q) = S(d_2^s, d_o) = 2A - CE(d_2^s||d_o) - CE(d_o||d_2^s) = 2A - (1 - \epsilon)CE(d_o||d_o) - \epsilon CE(d_2||d_o) + (1 - \epsilon)log((1 - \epsilon)^2 + \frac{\epsilon}{2}) + \epsilon log(\epsilon(1 - \epsilon) + \frac{\epsilon}{2}) = 2A - CE(d_o, d_o)$.

We now compute $CE(d_1||d_1), CE(d_1||d_o), CE(d_o||d_2), CE(d_o, d_o)$:

$CE(d_1||d_1) = -\epsilon log(\epsilon) - (1 - \epsilon)log(1 - \epsilon) = 0$.

$CE(d_1||d_o) = -\epsilon log(1 - \epsilon) - (1 - \epsilon)log(\epsilon) = -(1 - \epsilon)log(\epsilon) = -log(\epsilon)$.

$CE(d_o||d_2) = -(1 - \epsilon)log(0.5) - \epsilon log(0.5) = -log(0.5)$.

$CE(d_o||d_o) = -(1 - \epsilon)log(1 - \epsilon) - \epsilon log(\epsilon) = 0$.

We show that the value of the first advertiser is larger than that of the second advertiser:

$v_1(d_1^s) > v_2(d_2^s) \Leftrightarrow 2A - CE(d_1||d_1) > 2A - CE(d_o||d_2) + 0.5log(\frac{3\epsilon}{2} - \epsilon^2) \Leftrightarrow 2A > 2A + log(0.5) + 0.5log(\frac{3\epsilon}{2} - \epsilon^2) \Leftrightarrow 0 > log(0.5) + 0.5log(\frac{\epsilon}{2}) + 0.5log(3 - 2\epsilon) \Leftrightarrow -log(0.5) - 0.5log(3) > 0.5log(\frac{\epsilon}{2}) \Leftrightarrow \frac{8}{3} > \epsilon$.

We now turn to determine the winner. The prospect platform value for each advertiser is:

$PV(d_1^s, q, v_1(d_1^s)) = v_1(d_1^s) + U(d_1^s, q) = 4A + log(\epsilon) + log(2\epsilon(1 - \epsilon)) = 4A + log(\epsilon) + log(2) + log(\epsilon) + log(1 - \epsilon) = 4A + 2log(\epsilon) + log(2)$.

$PV(d_2^s, q, v_2(d_2^s)) = v_2(d_2^s) + U(d_2^s, q) = 4A + log(0.5) + 0.5log(\frac{3\epsilon}{2} - \epsilon^2) = 4A + log(0.5) + 0.5log(\epsilon) + 0.5log(1.5 - \epsilon) = 4A + log(0.5) + 0.5log(\epsilon) + 0.5log(1.5)$.

Thus, the winner is the second advertiser since:

$v_1(d_1^s) + U(d_1^s, q) < v_2(d_2^s) + U(d_2^s, q) \Leftrightarrow 4A + 2log(\epsilon) + log(2) < 4A + log(0.5) + 0.5log(\epsilon) + 0.5log(1.5) \Leftrightarrow 1.5log(\epsilon) < -log(2) + log(0.5) + 0.5log(1.5) \Leftrightarrow \epsilon < \frac{\sqrt[3]{6}}{4}$.

## A.2 Proof of Proposition 3

Recall that we use the same documents as in Proposition 2: $p_{d_o}(a) = 1 - \epsilon, p_{d_o}(b) = \epsilon, p_{d_1}(a) = \epsilon, p_{d_1}(b) = 1 - \epsilon, p_{d_2}(a) = 0.5, p_{d_2}(b) = 0.5$. Since $\lambda_1 = 1 - \epsilon, \lambda_2 = 0.5$, the unigram language models from

which the sponsored answers are sampled are: $p_{d_1^s}(a) = \lambda_1 p_{d_o}(a) + (1 - \lambda_1)p_{d_1}(a) = (1 - \epsilon)(1 - \epsilon) + \epsilon \cdot \epsilon = (1 - \epsilon)^2 + \epsilon^2$.

$p_{d_1^s}(b) = \lambda_1 p_{d_o}(b) + (1 - \lambda_1)p_{d_1}(b) = (1 - \epsilon)\epsilon + \epsilon(1 - \epsilon) = 2\epsilon(1 - \epsilon)$.

$p_{d_2^s}(a) = \lambda_2 p_{d_o}(a) + (1 - \lambda_2)p_{d_2}(a) = 0.5 \cdot (1 - \epsilon) + 0.5 \cdot 0.5 = 0.75 - \frac{\epsilon}{2}$.

$p_{d_2^s}(b) = \lambda_2 p_{d_o}(b) + (1 - \lambda_2)p_{d_2}(b) = 0.5 \cdot \epsilon + 0.5 \cdot 0.5 = \frac{\epsilon}{2} + 0.25$.

We compute $v_1(d_1^s), v_2(d_2^s), U(d_1^s, q), U(d_2^s, q)$ using again the linearity of $CE$ in its left argument:

$v_1(d_1^s) = S(d_1^s || d_1) = 2A - CE(d_1^s || d_1) - CE(d_1 || d_1^s) = 2A - (1 - \epsilon)CE(d_o || d_1) - \epsilon CE(d_1 || d_1) + \epsilon log((1 - \epsilon)^2 + \epsilon^2) + (1 - \epsilon)log(2\epsilon(1 - \epsilon)) = 2A - CE(d_o || d_1) + log(2\epsilon)$.

$v_2(d_2^s) = S(d_2^s, d_2) = 2A - CE(d_2^s || d_2) - CE(d_2 || d_2^s) = 2A - 0.5CE(d_o || d_2) - 0.5CE(d_2 || d_2) + 0.5 log(\frac{3}{4} - \frac{\epsilon}{2}) + 0.5 log(\frac{1}{4} + \frac{\epsilon}{2}) = 2A - 0.5CE(d_o || d_2) - 0.5CE(d_2 || d_2) + 0.5 log(\frac{3}{4}) + 0.5 log(\frac{1}{4})$.

$U(d_1^s, q) = S(d_1^s, d_o) = 2A - CE(d_1^s || d_o) - CE(d_o || d_1^s) = 2A - (1 - \epsilon)CE(d_o || d_o) - \epsilon CE(d_1 || d_o) + (1 - \epsilon)log((1 - \epsilon)^2 + \epsilon^2) + \epsilon log(2\epsilon(1 - \epsilon)) = 2A - CE(d_o || d_o)$.

$U(d_2^s, q) = S(d_2^s, d_o) = 2A - CE(d_2^s || d_o) - CE(d_o || d_2^s) = 2A - 0.5CE(d_o || d_o) - 0.5CE(d_2 || d_o) + (1 - \epsilon)log(\frac{3}{4} - \frac{\epsilon}{2}) + \epsilon log(\frac{1}{4} + \frac{\epsilon}{2}) = 2A - 0.5CE(d_o || d_o) - 0.5CE(d_2 || d_o) + log(\frac{3}{4})$.

We compute all the following $CE(\cdot || \cdot)$ terms:

$CE(d_1 || d_1), CE(d_1 || d_o), CE(d_o || d_1)CE(d_o || d_2)$, $CE(d_2 || d_o), CE(d_o, d_o), CE(d_2, d_2)$:

$CE(d_1 || d_1) = -\epsilon log(\epsilon) - (1 - \epsilon)log(1 - \epsilon) = 0$.

$CE(d_1 || d_o) = -\epsilon log(1 - \epsilon) - (1 - \epsilon)log(\epsilon) = -(1 - \epsilon)log(\epsilon) = -log(\epsilon)$.

$CE(d_o || d_1) = -(1 - \epsilon)log(\epsilon) - \epsilon log(1 - \epsilon) = -(1 - \epsilon)log(\epsilon) = -log(\epsilon)$.

$CE(d_o || d_2) = -(1 - \epsilon)log(0.5) - \epsilon log(0.5) = -log(0.5)$.

$CE(d_2 || d_o) = -0.5 log(1 - \epsilon) - 0.5 log(\epsilon) = -0.5 log(\epsilon)$.

$CE(d_o || d_o) = -(1 - \epsilon)log(1 - \epsilon) - \epsilon log(\epsilon) = 0$.

$CE(d_2 || d_2) = -0.5 log(0.5) - 0.5 log(0.5) = -log(0.5)$.

We show that the user utility given the sponsored answer of the first advertiser is larger than the user utility given the sponsored answer of the second advertiser:

$U(d_1^s, q) > U(d_2^s, q) \Leftrightarrow 2A - (1 - \epsilon)CE(d_o || d_o) - \epsilon CE(d_1 || d_o) + (1 - \epsilon)log((1 - \epsilon)^2 + \epsilon^2) + \epsilon log(2\epsilon(1 - \epsilon)) > 2A - 0.5CE(d_o || d_o) - 0.5CE(d_2 || d_o) + (1 - \epsilon)log(\frac{3}{4} - \frac{\epsilon}{2}) + \epsilon log(\frac{1}{4} + \frac{\epsilon}{2}) \Leftrightarrow 2A - CE(d_o || d_o) > 2A - 0.5CE(d_o || d_o) - 0.5CE(d_2 || d_o) + log(\frac{3}{4}) \Leftrightarrow 0 > 0.25 log(\epsilon) + log(\frac{3}{4}) \Leftrightarrow \epsilon < \frac{4^4}{3^4}$.

We compute the prospect platform value for both advertisers in order to determine who is the winner:

$PV(d_1^s, q, v_1(d_1^s)) = v_1(d_1^s) + U(d_1^s, q) = 2A - (1 - \epsilon)CE(d_o || d_1) - \epsilon CE(d_1 || d_1) + \epsilon log((1 - \epsilon)^2 + \epsilon^2) + (1 - \epsilon)log(2\epsilon(1 - \epsilon)) + 2A - (1 - \epsilon)CE(d_o || d_o) - \epsilon CE(d_1 || d_o) + (1 - \epsilon)log((1 - \epsilon)^2 + \epsilon^2) + \epsilon log(2\epsilon(1 - \epsilon)) = 2A - CE(d_o || d_1) + log(2\epsilon) + 2A - CE(d_o || d_o) = 4A + log(\epsilon) + log(2\epsilon)$.

$PV(d_2^s, q, v_2(d_2^s)) = v_2(d_2^s) + U(d_2^s, q) = 2A - 0.5CE(d_o || d_2) - 0.5CE(d_2 || d_2) + 0.5 log(\frac{3}{4} - \frac{\epsilon}{2}) + 0.5 log(\frac{1}{4} + \frac{\epsilon}{2}) + 2A - 0.5CE(d_o || d_2) - 0.5CE(d_2 || d_o) + (1 - \epsilon)log(\frac{3}{4} - \frac{\epsilon}{2}) + \epsilon log(\frac{1}{4} + \frac{\epsilon}{2}) = 4A + 0.5 log(0.5) + 0.5 log(0.5) + 0.5 log(\frac{3}{4}) + 0.5 log(\frac{1}{4}) + 0.25 log(\epsilon) + log(\frac{3}{4}) = 4A + log(\frac{3\sqrt{3}}{32}) + 0.25 log(\epsilon)$.

The winner is again advertiser $i = 2$, since she maximizes the prospect platform value:

$v_1(d_1^s) + U(d_1^s, q) < v_2(d_2^s) + U(d_2^s, q) \Leftrightarrow 4A + log(\epsilon) + log(2\epsilon) < 4A + log(\frac{3\sqrt{3}}{32}) + 0.25 log(\epsilon) \Leftrightarrow 2 log(\epsilon) + log(2) < log(\frac{3\sqrt{3}}{32}) + 0.25 log(\epsilon) \Leftrightarrow 1.75 log(\epsilon) < log(\frac{3\sqrt{3}}{32}) - log(2) \Leftrightarrow \epsilon < 0.23$.