# OpenReview forum: "Sponsored Question Answering"
_ACM.org/SIGIR/ICTIR/2024/Conference — ICTIR 2024_

### Official Review · Reviewer_6RZY · 2024-05-17

**Rating:** -2
**Confidence:** 4

**Objective Part Of Review:**

The authors introduce a formal analysis of a sponsored QA platform, which combines an organic answer with an advertisement to generate a sponsored answer. Advertisers bid on these sponsored answers, and the platform, inspired by Generalized Second Price Auctions (GSPs), selects the winning advertiser, determines their payment, and displays the sponsored answer to the user. Their analysis reveals several key findings, including the incentivization of truthful bidding by advertisers and the stability of the resultant setting with characteristics akin to VCG auctions.

**Subjective Part Of Review:**

I find the work very interesting as a discussion and analysis
However, I find it difficult to relate to real-life aspects due to lack of implementation and evaluation

Moreover, the theory part presented is not motivated much
It is not clear why different definitions and lemmas needed
at some point one may question, whether we are doing any better than already existing methods like VCG

having said this, I found the idea of social welfare very interesting

---

### Official Review · Reviewer_PMRg · 2024-05-18

**Rating:** -1
**Confidence:** 3

**Objective Part Of Review:**

The paper makes a first step towards a formal analysis of a sponsored QA platform, where ads are combined with organic anwers using LLMs, using a game-theorietical approach. The problem is formalized using unigram language models, which is a reasonable starting point.

The main findings are that "auction winner is not necessarily that with the highest bid" and that "the attained user utility is not the maximal possible with respect to selecting other advertisers".
First, it remains unclear what these findings mean for advertisers and the QA platform. If the attained user utility is not maximal, how much is getting "lost" there? This could be analyzed, e.g., using some simulations. It also remains unclear whether the QA platform should do anything differently.
Second, it remains an open question to what extent these findings hold for auto-regressive text generation approaches.

Overall, the work represents an interesting start to a relevant problem and the formal analysis appears to be sound. However, the findings remain purely theoretical and hold only in a highly simplified setting, and it remains unclear if they have any practical implications. Either some simulation studies on the simplified setting or more formal work hinting how the approach could be extended to auto-regressive LLMs would be needed. In my view, the current contributions are insufficient to warrant a paper.

**Subjective Part Of Review:**

The approach appears to be sound, but I admit that I haven't checked in detail all the proofs.

- The introduction and related work seems to mix a couple of different types of systems, including search engines, conversational agents, and QA platforms. It might be better to simply focus on a QA platform throughout, pointing out that sponsored search is relevant across all these systems.

- The notion of social welfare could perhaps be better explained. As I understand this is something that the QA platform optimizes for, but it is actually up to the platform how they balance the trade-off between advertiser value and user utility. Currently, these two seem to be considered with equal weights. If the platform chose, say, to place more emphasis on satisfying advertisers, would that change the findings?

---

### Official Review · Reviewer_6goK · 2024-05-19

**Rating:** 2
**Confidence:** 3

**Objective Part Of Review:**

An interesting and timely contribution that considers how advertisements could be auctioned in the context of chat conversations / Q&A with LLMs.

The written text does state eventually in paragraph how GSP and VCG are the same when a single ad is selected, but it would be good for a less knowledgeable reader to already make that clear in the intro.

This reviewer is not an expert in auction design for advertising. The text is generally clear, but in the case of the ad-agnostic approach, it would be good to state explicitly how the results differ from advertising in search. Are design considerations and common solution the same, and is it just a matter of notation/application of what we know from search to QA, or are we covering new ground already in this part of the paper?

Again, a point with clarity of writing, is that the importance of including the answer generation model into the auctioning process is not clearly stated. It is motivated a little in section 4, but the motivation could be strengthened. Since the key questions seem to be that the d_i^S should still be similar to both d_0 and d_i, I wondered why an NLI or textual entailment model would not be considered here (say to "implement" eq 4). I thought the finding in the text right below eq. 6 was kind-a obvious, but perhaps good to state nevertheless - maybe make even more explicit that it is the role of similarity that leads to this finding.

Finally, with regard to related work, the authors seem to have missed recent publications by the WebIS team on inserting advertisements in chat, and whether people can spot these (see PhD research by Ines Zelch).

**Subjective Part Of Review:**

I quite liked the paper, but am not an expert in advertising for online services, so it could just be novelty for me. The authors could do a better job at situating their contribution in the literature (the related work is quite good though - so it may also just be my superficial degree of knowledge from this research niche).

---

### Official Review · Reviewer_pi1S · 2024-05-22

**Rating:** 1
**Confidence:** 4

**Objective Part Of Review:**

The paper investigates sponsored QA responses.

There are some strengths:

- The general topic of sponsored vs organic results in emerging chat and RAG based search engine  SERPs is very timely and relevant.

- While the analysis is perhaps "a first step" is helps to clear up several of the relevant aspects to consider in such systems.

- There is an interesting sketch of sponsored answer generation.

There are some limitations:

- The SERP/UI both in its restrictions (otherwise displaying sponsored results is no issue) as well as in its opportunities.  E.g., will an organic result be replaced by a sponsored one, and how will this be labeled?  Will an organic result (e.g. the book example) be expanded by a sponsored follow-up action (e.g., "and you can buy it XXX")?   Or ...  This feels relevant as this leads to assumptions on the auction part (is this still the question/query?  or a query+response-type such as "book", or query+exact response, or ...

- There is limited attention on the advertisers side of the problem in a more practical sense (now a variant of a vickery second prize account bidding on query words): how would this extend to q+d_0? would this result in an optimal prize given the utility to the advertiser?

- It feels unavoidable to discuss the ethics and desirability of this type of proposal, at least a reflection on the importance of addressing these aspects.

**Subjective Part Of Review:**

Minor:

- A detailed (running) example would greatly help digest the content of the paper...

---

### Meta-Review · Area_Chair_FNbK · 2024-05-30

**Recommendation:** Accept (Oral)
**Confidence:** 3

**Metareview:**

Motivated by the move away from search toward question answering, the submission models and theoretically analyzes a platform for sponsored question answering. The proposed platform fuses an organic answer with an advertisement using, for instance, a unigram language model or a LLM. The model considers platform value, advertiser value as well as user utility. Advertisers bid on a fused answer based on a generalized second-price auction mechanism. One key finding of the submission is that a dominant strategy for advertisers is to bid based on their true value, which avoids effects such as shading. In a subsequent analysis, the submission considers a concrete fusion mechanism, namely a unigram language model which linearly interpolates token probabilities estimated from the organic answer and the ad. An interesting finding here --under the assumed fusion mechanism-- is that the winning advertiser is not necessarily the one having the maximal value.

All reviewers agreed that the submission addresses an interesting and timely problem. It is clear from their reviews, that the reviewers regard the submission as a first formal step with potential for follow-up work and analyses. They also pointed out several issues, which the authors should address. This includes the ethics and desirability of such a platform in general, several fixable presentations issues, and a more complete discussion of related work.